# DNA Methylation Malleability and Dysregulation in Cancer Progression: Understanding the Role of PARP1

**DOI:** 10.3390/biom12030417

**Published:** 2022-03-08

**Authors:** Rakesh Srivastava, Niraj Lodhi

**Affiliations:** 1Molecular Biology and Biotechnology Division, CSIR-National Botanical Research Institute, Lucknow 226001, India; raakeshshrivastav@yahoo.com; 2Clinical Research (Research and Development Division) Mirna Analytics LLC, Harlem Bio-Space, New York, NY 10027, USA

**Keywords:** DNA demethylases, DNA demethylases inhibitors, PARP1, poly(ADP-ribose), DNA methylation, tumor suppressor gene, oncogene, tumor progression, cancer cells

## Abstract

Mammalian genomic DNA methylation represents a key epigenetic modification and its dynamic regulation that fine-tunes the gene expression of multiple pathways during development. It maintains the gene expression of one generation of cells; particularly, the mitotic inheritance of gene-expression patterns makes it the key governing mechanism of epigenetic change to the next generation of cells. Convincing evidence from recent discoveries suggests that the dynamic regulation of DNA methylation is accomplished by the enzymatic action of TET dioxygenase, which oxidizes the methyl group of cytosine and activates transcription. As a result of aberrant DNA modifications, genes are improperly activated or inhibited in the inappropriate cellular context, contributing to a plethora of inheritable diseases, including cancer. We outline recent advancements in understanding how DNA modifications contribute to tumor suppressor gene silencing or oncogenic-gene stimulation, as well as dysregulation of DNA methylation in cancer progression. In addition, we emphasize the function of PARP1 enzymatic activity or inhibition in the maintenance of DNA methylation dysregulation. In the context of cancer remediation, the impact of DNA methylation and PARP1 pharmacological inhibitors, and their relevance as a combination therapy are highlighted.

## 1. Introduction

Epigenetic processes are functional chromatin alterations that occur as a result of heritable changes in the genes or genome that are not caused by changes in the nucleotide sequences. These precise epigenetic markers undergo dynamic alterations during development and cellular differentiation, which eventually aid in the maintenance and generation of diverse types of cells in an organism [1]. Epigenetic processes play a part in the various phases of cell differentiation as set out by the precursor or primary cells; cells have a DNA sequence similar to that of primary cells, which also give them long-term cellular memory for cell differentiation. Epigenetic modifications are likely to have a substantial influence on the onset and progression of many diseases. Epigenetics has added unique insights to disease traits that cannot be explained by genetic or environmental causes, enriching human disease knowledge. Epigenetic changes are being utilized to understand several basic features of complex diseases, such as late-onset of and variations in disease symptoms [2,3].

The dynamics of DNA methylation are an important epigenetic signature that has been widely researched among epigenetic processes. In recent years, DNA methylation and its dynamic control have been extensively integrated into modern epigenetic encoding models. In mammals, epigenetic modifications of DNA predominantly involve the addition of a methyl group of the cytosine base to carbon five before guanine, subsequently generating 5-methylcytosine (5mC). In mammals, the majority of DNA methylation materializes in the background of the CpG-dinucleotide framework (characterized by cytosine, guanine and the phosphate group between them). CpG islands are high-density CpG-dinucleotide units found in interspersed areas, primarily in the promoter and regulatory regions [4,5,6,7,8]. The human genome has about 29 million CpG sites, with nearly 60–80% of them being methylated in normal somatic cells [9]. Surprisingly, CpG sites are not uniformly distributed throughout the genome; in contrast, the majority of the genome is devoid of CpG sites, with just one-fifth of the predicted ratio of CpG dinucleotides being present [8]. The majority of CpG islands often span gene promoters and housekeeping genes and are 500–1000-base-pair (bp) long [10]. Significantly, DNA methylation occurs in 70% of all CpG dinucleotides and 40% of genes with CpG-rich islands in the genome [5,11]. Numerous methyl-binding proteins recognize methylated CpGs as binding sites that participate in the recruitment of chromatin-remodeling protein or machinery, thereby facilitating gene silencing and inactivation, and chromatin condensation [8,12,13,14]. Methylation on non-CpG regions has also been described to be affecting DNA–protein interactions, chromatin structure and stability, and gene regulation [15]. Non-CpG methylation (CpH, where H = A, C, or T) has also been reported in oocytes, human embryonic stem cells and neurons [16,17,18,19,20].

For dynamic chromatin modification, the connection among gene regulation, histone modification and DNA methylation is highly coordinated and synchronized [21,22,23,24]. Poly(ADP-ribosyl)ation is a catalytic activation of PARP1 that occurs when it catalyzes the addition of ADP-ribose (ADPr) to a pre-existing chain of poly(ADPr) of target proteins, including itself via auto-poly(ADP-ribosyl)ation [25,26]. The activity of PARP1 is engaged in several biological and cellular functions, including histone or chromatin alterations and consequent gene expression modulation. However, the epigenetic mechanism of PARP1 differs from that of DNA methylation because it also enzymatically opens condensed chromatin in advance of transcriptional activity [27,28]. The current review expands on PARP1’s potential importance as a new therapeutic target for clinical applications by expanding its numerous roles linked with DNA methylation in normal and cancer cells.

## 2. Dynamic Behavior of DNA Methylation and Demethylation

In mammals, cytosine methylation is mostly limited to the symmetrical CpG framework [19,29]. To inactivate transcription, methylation is most commonly found in the CpG islands of target-gene promoter–exon regions. Particularly, CpG islands are found in the promoters of half of all genes [30]. In contrast to CpG-island promoters and shores, gene bodies tend to have a considerate amount of 5mC, which corresponds to active gene expression [10] (Figure 1). CpG-deficient regulatory areas, such as tissue-specific enhancers, are classified as lowly methylated regions, with average DNA-methylation frequencies varying from 10 to 50% [31] (Figure 1). At verified promoter–enhancer pairings, DNA methylation levels in the enhancer region have been linked to gene activity, with a low level of 5mC indicating greater gene expression [10,32]. CpG dinucleotides have a low incidence in the human genome, but they are interspersed with sections with high numbers of these sequences that are linked to gene regulatory regions [33]. Particularly, CpG sites are usually methylated across the genome, although sites within CpG islands are not methylated unless the corresponding gene is silenced. DNA methylation has been connected to non-reversible events, including imprinting, dosage compensation, or the silencing of developmentally regulated genes with cell differentiation, as well as a potentially damaging transposon and virally inserted sequences [8,34].

During gametogenesis and shortly after fertilization, DNMT3L works as an accessory protein for DNMT3A- and DNMT3B-mediated de novo DNA methylation [47]. Interestingly, only germ cells and embryonic stem cells express DNMT3L; however, no such functions have been reported in somatic cells [48]. DNMT3L, together with DNMT3A and DNMT3b, is primarily expressed in the postnatal female germline for the development of DNA-methylation patterns [49]. DNMT3L is implicated in the regulation of repetitive elements, as well as germ-cell imprinting [47,50]. According to the study, DNMT3L has two roles in the differentiation of embryonic stem (ES) cells; firstly, it works as a positive controller of DNA methylation in the housekeeping-gene part and, secondly, it functions as a negative controller of DNA methylation at bivalent gene promoters [51]. Notably, DNMT2 (also called tRNA aspartic acid methyltransferase 1 (TRDMT1)) exhibits weak activity of methyltransferase in vitro and its removal has an insignificant influence on CpG methylation levels and no apparent effects on developmental phenotypes [52,53]. On the other hand, the anticodon loop of aspartic-acid transfer RNA is methylated by DNMT2 [54]. Based on genetic factors, DNMT2/TRDMT1-dependent RNA modifications are important in determining the coding signature of sperm small non-coding RNA, which is required for paternal epigenetic memory and in the transmission of paternally acquired metabolic diseases to offspring [44,55].

In comparison to most histone modifications, DNA methylation is rather stable. However, DNA demethylation (the lack of DNA methylation) has been seen in a variety of biological and developmental contexts. Demethylation changes to DNA can occur in two ways, active or passive pathway [56]. In the active DNA demethylation process, the enzymatic removal of or alteration in the methyl group of 5mC takes place [57]. Passive DNA demethylation, on the other hand, occurs as a result of the loss of maintenance methylation [44]. Passive DNA demethylation happens during multiple cycles of replication due to the absence of efficiently working DNA-methylation maintenance machinery, for instance, DNMT1 suppression or absence of the DNA hypomethylation effect. Passive DNA demethylation can also occur during mammalian development, particularly during pre-implantation development in the maternal genome [44,58,59]. 5mC demethylation to produce 5-hydroxymethylcytosine (5hmC) commonly primarily involves 5mC oxidation through ten–eleven translocation (TET) methyl cytosine dioxygenases (Figure 2B and Figure 3). Further, TET enzymes hydroxylate 5hmC to produce 5-formylcytosine (5fC) and 5-carboxylcytosine (5caC) in a series of steps. Thymine DNA Glycosylase (TDG) recognizes the intermediary bases 5fC and 5caC and disrupts the glycosidic bond, leading to an apyrimidinic site. After that, the excision at the base is repaired. In different oxidative deamination mechanisms, 5hmC can be oxidatively deaminated by AID (activation-induced cytidine deaminase)/APOBEC (apolipoprotein B mRNA editing enzyme catalytic subunit) deaminases to produce 5-hydroxymethyluracil (5hmU). 5mC can also be transformed into thymine by activation-induced deaminase (AID) or apolipoprotein B RNA-editing catalytic components 2b and 2a (Apobec2b, 2a). Methyl-CpG Binding Domain 4 (MBD4), Nei Like DNA Glycosylase 1 (NEIL1), TDG and Single-Strand-Selective Monofunctional Uracil-DNA Glycosylase 1 (SMUG1) can all cleave 5hmU. Base-excision repair (BER) enzymes subsequently repair the apyrimidinic site and T: G mismatches to generate cytosine. The TET family most frequently demethylates DNA by dioxygenases [60].

Different TET gene isoforms are expressed in various cells and organs. In the TET dioxygenase family, a minimum of two TET1, one TET2 and three TET3 isoforms has been reported (Figure 2B) [56,61]. Embryonic stem cells, initial embryo stages and primordial germ cells appear to be the only places where the full-length canonical TET1 isoform is found. The dominant TET1 isoform in most somatic tissues, notably in mice, derives from the activation of an alternative promoter, resulting in a truncated transcript and a smaller protein known as TET1s [62]. TET3 isoforms also include full-length form, TET3FL; a short-form splice form, TET3s; and another variant reported in neurons and oocytes, referred to as TET3o. TET3o is generated by a distinct promoter and contains a unique initial N-terminal exon, which encodes for 11 amino acid residues. There has been no such report yet that TET3o can be found in embryonic stem cells, other cell types, or adult animal tissue. *TET1* expression is scarcely detectable in zygotes and oocytes, while *TET2* has a low level of expression and the *TET3* variation *TET3o* is almost nonexistent at the two-cell stage. When exceptionally large-scale rapid demethylations occur in neurons, oocytes and zygotes at the one-cell stage, TET3o could be the most common TET enzyme used [56,63,64,65,66,67].

## 3. DNA Methylation and Demethylation in Cancer Progression

Epigenetics is widely documented to play a role in cancer development; a significantly changed epigenome, such as aberrant DNA methylation and histone modification configurations, is now thought to be a typical cancer signature and hallmark. Recent progress has provided the mechanistic insight of DNA methylation–demethylation dynamics, as well as their prospective regulatory roles in cellular differentiation and carcinogenic progression [10,68,69] The phenomena of DNA hypermethylation and tumor-suppressor-gene (TSG) silencing have attracted the most interest in cancer progression. Hypermethylation can, in principle, play a critical role in cancer development and progression. Furthermore, hypomethylation is gradually being recognized as a promising pathway for cancer prometastatic gene activation. The malfunctioning of methylation machinery or of DNMT enzymes has been attributed to the abnormal DNA methylation topography in cancer cells. The finding of 5hmC, 5fC and 5caC, on the other hand, has anticipated that a failure of the demethylation enzymatic system could result in DNA methylation marks asymmetry and reprogramming [70].

Chemical carcinogens or pathological factors can cause genetic mutations that affect DNMT functions or expression levels, resulting in genome-wide methylation profile variations and cancer-stimulating gene-expression alteration, such as reducing TSG expression while boosting genomic instability and oncogene expression [68,69,71,72]. Accumulating studies have revealed that gene expression anomalies produced by DNMT activity and function are linked to the incidence and progression of many malignancies (Appendix A). Hypermethylation and hypomethylation are thought to be separate mechanisms in cancer that target various programs at different stages of carcinogenesis. Many malignancies exhibit genome-wide hypomethylation and promoter hypermethylation, which are linked to carcinogenesis. Hypomethylation across the genome has been linked to an increase in genomic instability [73,74]. Hypermethylation of CpG islands in gene promoters, on the other hand, can inhibit TSGs and affect crucial physiological functions, including apoptosis, angiogenesis, cell cycle, cell adhesion and DNA repair [75]. Notably, it has been reported that TET protein expression or function is frequently dysregulated in a variety of malignancies. In vivo, TET deficiency is significantly connected to the start and progression of hematologic malignancies (Appendix A). Many forms of malignancies are linked to TET impairments, TET loss-of-function alterations and TET loss of function produced by hypoxia and other regulatory and metabolic disturbances [76].

## 4. Poly(ADP-ry)lation of DNMT1 Determines DNA Methylation

PARP1 is a multifunctional-domain protein (1014 amino acids, 113 kDa); its N-terminal domain (1–353 AA) contains three DNA-binding domains (zinc fingers ZF1, ZF2 and ZF3) and a nuclear-localization-sequence domain (NLS) (Figure 2C). ZF1 and ZF2 recognize and bind to damaged DNA sites, while the function of ZF3 is to activate enzymes and NLS leads newly translated PARP1 to the nucleus. The central automodification domain (389–643 AA) is an auto-poly(ADP-ribosyl)ation site for PARP1 and functionally very important, composed of BRCT (which mediates protein–protein interactions) and WGR (which interacts with ZF1, ZF3 and catalytic domains) domains. The C-terminal catalytic domain (662–1014 AA) is composed of the NAD acceptor site to perform poly(ADP-ribosyl)ation enzymatic activity [77].

PARP1 was originally known as a DNA-repair enzyme, as until recent years, other functions were not known. Now, we know PARP1 controls the transcription regulation [78], NF-κB-dependent immune response [79], ribosome biogenesis, epigenetic inheritance of mechanism of gene expression through mitotic bookmarking [80,81,82] and differential DNA methylation [83,84]. PARP1 is a protein that catalyzes the transfer of ADP-ribose units from NAD^+^ to specific target proteins and controls important physiological processes such as DNA methylation, DNA damage response, chromatin remodeling and gene expression. This process, known as poly(ADP-ribosyl)ation, produces one ADP-ribose and one nicotinamide for every NAD^+^ molecule processed. The ADP-ribose unit is subsequently connected to carboxyl groups in the target protein structure by glutamate, aspartate, lysine, arginine and serine residues [27,85]. When poly(ADP-ribose) (PAR) accumulates, it features a strong negatively charged nucleic-acid-like structure [86] and neutralizes positively charged groups, mediating chromatin de-condensation and stimulating transcription [81]. Among all the PARP1 functions, DNA methylation is not well understood yet. During carcinogenesis, major DNA methylation change happens globally and certain genes are targets of aberrant methylation, causing the epigenetic silencing of TSGs (Figure 3). However, a plausible mechanism published by [87] is that, after auto poly(ADP-ribosyl)ation of PARP1, poly(ADP-ribosyl)ated covalent chains recruit DNMT1 and block its catalytic activity, thus preventing aberrant hypermethylation. The mechanisms of DNMT1 recruitment and PARP1 activation at CpG islands remain unknown. Another study has later confirmed that PARP1 could directly impact DNA methylation patterns governing DNMT1 transcription and activity in mouse primordial germ cells via poly(ADP-ribosyl)ation [88]. Not only PARP1 poly(ADP ribosyl)ates DNMT1 to prevent its function and maintenance of methylation on newly formed DNA after replication [87], but the auto-poly(ribosyl)ation of PARP1 also facilitates *DNMT1* expression by loosening chromatin or moving away from the DNA and providing access to transcription machinery [89]. In fact, there is a number of reports which shows that PARP1 binding to DNA sequences prevents DNA methylation and poly(ADP-ribosyl)ation has been indeed revealed to maintain the unmethylated status of regulatory zones of particular genes, including *DNMT1*, *p16* (also known as p16INK4a, cyclin-dependent kinase inhibitor 2A)*, SMA* (smooth muscle actin)*, THBD* (thrombomodulin)*, TET1*, the *DMR1* (differentially methylated region 1) imprinted region, as well as certain other pluripotency-associated genes [89,90,91,92,93,94,95,96,97].

The epigenetic function of PARP1 has been revealed that it bookmarks the promoter of cell-identity genes during mitosis, which is crucial for expression of genes to survive daughter-cells survival when they enter into G1 [81,82]. It is possible, sequences of these promoters that remain to protect from methylation due to the presence of PARP1. In cancer cells, aberrant poly(ribosyl)ation activity or its controlling PARG activity disrupts DNA methylation; in some cases, it may be hyper- or hypomethylation depending on the cues that affect the poly(ribosyl)ation of DNMT1 [98]. Poly(ADP ribosyl)ation is reversible PARP1 enzymatic activity and PARG quickly maintains the homeostasis in the cell by degrading the PAR chain that prevents the disastrous effect on the cell. In PARG or homologous DNA-repair-defective cells, the enzymatic activity of PARP1 is increased and imbalances NAD homeostasis [99,100]. In case of excessive enzymatic activity, PARP1 poly(ADP ribosyl)ates itself and interacts with DNMT1 to form a complex and prevent DNMT1 functional activity to methylate DNA [87,89,101]. It leads to a scarcity of NAD^+^ levels in cancer cells. It is a well-known fact that excessive PARP enzymatic activation causes a reduction in NAD^+^ levels [102]. NAD^+^ levels may decline to 20–30% of their previous levels under such circumstances, putting a rate limitation on the sirtuin enzymes [103]. The enzymatic activities of histone deacetylase (Sirtuins) largely reduce in these conditions because Sirtuins compete for NAD^+^ with PARP1 [104,105,106] and this is followed by decreased *SIRT1* expression [105]. Moreover, it may facilitate the transcription initiation of some oncogenes (responsible for cancer progression) by being unable to remove histone acetylation from already unmethylated genes (Figure 4). In this case, the use of the PARP1 inhibitor reverts all the functions to suppress the expression of oncogenes [102]. An increase in PARP1 enzymatic activity leads to inactivate DNMT1 enzymatically and causes hypomethylation on the promoter or downstream region of genes in normal cells (Figure 5).

The silencing of the *DNMT1* gene may be responsible for the global loss of methylation [87]. However, it opens other pathways to suppress genes by spreading heterochromatin (next section). DNA methylation dynamically changes in response to environmental cues, whereby DNA damages lead to the activation of DNA repair machinery and enzymatic activity of repair enzymes, including PARP1. PARP1 controls gene expression in two ways, i.e., a) by binding to the upstream or downstream gene and b) by performing the poly(ADP-ribosyl)ation of genes. These two pathways are contrasting; PARP1 provides access to transcription machinery after auto-poly(ADP-ribosyl)ation by leaving the binding sites of genes, such that genes become transcriptionally active. PARP1 localizes the DNMT1 promoter in normal cells [89], perhaps identifying and protecting unmethylated regions in the genome from methylation, thus contributing to the epigenetic control of gene expression [87].

## 5. PARP1 in DNA Hypermethylation and Its Effect on Cancer Progression

TSGs are required for proper cell development because they halt cell division, correct DNA errors and regulate programmed cell death. TSGs that do not act correctly can cause cells to develop out of control, leading to cancer. Studies on retinoblastoma, a rare childhood eye tumor, have led to the discovery of the first TSG [107]. The cell needs of their respective pathways determine how these TSGs are expressed. Cancer cells, in particular, lose full control of all genes, including TSGs. TSG function is reduced in all malignancies by a variety of mechanisms, one of which is excessive DNA methylation. TET1 restores normalcy by reversing methylation. Excessive poly(ribosyl)ation activity in cancer cells poly(ADP-ribosyl)ates DNMT1 and renders its activity; TET1 is similarly poly(ADP ribosyl)ated, which is one of the causes of hypermethylation of DNA in these cells (Figure 6).

### 5.1. Effect of DNA Hypermethylation on TSG (P53 and NF-κB) Expression

P53, one of the major tumor suppressor proteins, and its loss of function by mutations or loss of expression cause more than 50% of human cancers. P53 also plays a key role in a multitude of DNA-damage response pathways [86]. It has been reported in several papers that P53 and PARP1 interact at multiple levels [108]. P53 is not only a covalent poly(ADP-ribosyl)ation target [109,110], but it also possesses a high-affinity non-covalent association with poly(ADP-ribosyl) [111]. Dysregulated poly(ribosyl)ation activity in cancer cells could be one of the possibilities to downregulate P53 expression via DNA hypermethylation on its gene region.NF-κB, the master regulator, mediates the crosstalk between cancer and inflammation at multiple levels. Enhanced NF-κB function can cause pro-inflammatory cytokine production in tumor tissues, which significantly contributes to the pro-tumorigenic microenvironment [112].

In short, RelA/p65, RelB, c-Rel, p50 (NF-κB1) and p52 (NF-κB2) are members of the NF-κB transcription-factor family which exist as homo- and hetero-dimers (normally, NF-κB, p50 and p65) [113]. The composition of the dimer influences NF-κB enzyme stimulation, DNA-binding efficiency and DNA-sequence preference. In the absence of a signal, inhibitory proteins (IκBα, β or ε) interact with the dimers of NF-κB and segregate them in the cytoplasm [114]. The activation of the pathway causes the proteasomal degradation of inhibitors (IκBα, β or ε) [115], letting the NF-κB dimer enter the nucleus and trigger genes accountable for targeting cancer cells and pro-inflammatory transcription programs [116]. In response to a signal, PARP1 acts as a unique and essential transcriptional coactivator of NF-κB in vivo. PARP1’s coactivator action is dependent on direct protein–protein interactions with both NF-κB subunits; extending PARP1 enzymatic activity plays a major and unique canonical transcriptional coactivatory role for NF-κB-dependent gene regulation [117,118]. In AML cells, the NF-κB pathway is constitutively activated [119]. By binding to the promoter region of the *PARP1* gene and regulating *PARP1* gene transcription, RelA/p65 promotes DNA repair. PARP1 depletion decreases NF-κB function, suggesting that NF-κB and PARP1 form a DNA-repair positive feedback loop [120].

### 5.2. Control of DNA Hypermethylation

#### DNA Methyltransferases Inhibitor

Because TSG hypermethylation is a hallmark feature, much effort has gone into finding medicines that induce the DNA demethylation of these genes to restore their expression and function in cancer cells. Growing evidence suggests that inhibiting DNMTs is associated with decreased tumorigenicity and increased expression of TSGs. As a result, DNMTs are regarded as promising drug candidates for particular anti-cancer treatments. Nucleoside analogs decitabine, 5-Azacytidine and zebularine are the three most often used DNMT catalytic inhibitors, each with a distinct mode of action, discussed below. Several other subsequent generations of drugs have been discovered and checked for their role in cancer progression (Table 1).

1.Decitabine

In recent progress, DNA methylation has revealed its importance in the development of malignancies; this attracted attention, from the chemotherapeutic perspective, on the use of 5-Aza-2′-deoxycytidine (5-AZA-CdR, decitabine) for cancer treatment [121]. Decitabine (a DNA methyltransferase (DNMT) inhibitor approved by the FDA) is useful for the reduction in hematological neoplasms [122]. It transforms phosphorylated nucleotides into active forms, integrates DNA as a cytosine replacement, irreversibly binds to DNMTs and confines enzymes on DNA. As a result, the DNMT pool is depleted and its function is inhibited [58,122]. Decitabine at high micromolar dosages promptly causes DNA damage and cytotoxicity [123,124]. As previously stated, one of the most persistent signatures of malignancies is DNA hypermethylation on TSGs [125]. It leads to the loss of the regulated expression of TSGs that facilitates cancer cell growth [126,127,128].

Decitabine, an epigenetic drug that inhibits DNA methylation and has been licensed by the FDA, is being used to cure myelodysplastic syndrome (MDS). Several studies are in progress to treat acute myeloid leukemia (AML) and other malignancies. It helps to restore TSGs that have been silenced by abnormal DNA methylation, which is prevalent in all different cancers. It also suppresses *DNMT3B* expression, a de novo DNA methylating enzyme [129,130]. AML and MDS have been linked to DNMT3A mutations [131,132]. Both AML and MDS patients with these DNMT3A mutations have an unfavorable prognosis [133]. TNBC (triple-negative breast cancer) is a complex disease with poor survival. TNBC tumors have a lot of epigenetic biomarker genes with hypermethylated promoters. Decitabine treatment sensitizes TNBC cells that could be used for second-line treatment of chemoresistant patients [122,134]. However, only around 40% of Decitabine-treated AML patients eventually gain benefit from it and, even among responders, recurrence is common; these cells probably develop drug resistance due to adaptive pyrimidine-metabolism-network reactions [135].

2.5-Azacytidine

5-Azacytidine is a cytosine analog that has been found to promote DNA demethylation and is a powerful DNA methyltransferase inhibitor. Patients with higher-risk myelodysplastic syndrome (MDS) are treated with 5-azacytidine (Vidaza; Celgene Corporation, Boudry, Switzerland) [136]; it is also used for a subgroup of acute myeloid leukemia (AML) [137] and chronic myelomonocytic leukemia (CMML) patients [138]. By suppressing pancreatic-ductal-adenocarcinoma (PDAC) development in vivo, epigenetic reprogramming with 5-azacytidine induces an anti-cancer strategy in PDAC cells [139]. The de novo DNA methylating enzyme, DNMT3B, has been revealed to be inhibited by 5-azacytidine [129].

3.Zebularine

Zebularine is a nucleoside analog that, unlike 5-azaC, is chemically stable and orally accessible. However, it can resolve Aza’s shortcomings, including its cytotoxicity, instability and short half-life. Zebularine has been shown to inhibit DNMT in a variety of tissues in vitro and in vivo, including breast cancer, colorectal cancer, lung cancer and prostate cancer [140,141,142,143,144]. Zebularine can make tumor cells more chemosensitive and radiosensitive. Zebularine also possesses antimitotic and vascular inhibitory properties. For example, it stimulates the production of E-cadherin, a cellular gene that is typically suppressed by hypermethylation in malignancies [145]. Zebularine, in particular, has been shown to reactivate the silent *p16* gene and demethylate its promoter region in T24 bladder cancer cells [146].

**Table 1 biomolecules-12-00417-t001:** Recent update on DNMT inhibitors relatively to the regulation of the different types of cancer progression.

DNMT Inhibitor	Effect on Cancer Progression	References
Decitabine	Lung Cancer, Colorectal Cancer, Breast Cancer, Prostate Cancer, Liver Cancer, Acute Myeloid Leukemia	[147,148,149,150,151,152]
5-Azacitidine	Gastric Cancer, Acute Myeloid Leukemia, Germ-Cell Tumor, Esophageal Cancer, Colon Cancer	[153,154,155,156,157,158,159]
Zebularine	Colon Cancer, Liver Cancer, Pancreatic Cancer, Prostate Cancer, Medulloblastoma	[140,141,142,143,144]
Guadecitabine (SGI-110)	Germ-Cell Tumor, Ovarian Cancer, Liver Cancer, Urothelial Cancer	[160,161,162,163]
5-Fluro-2′ deoxycytidine	Urothelial Cancer, Colon Cancer	[164,165]
5,6, dihydro 5 azacytidine	T-Cell Acute Lymphocytic Leukemia, Acute Myeloid Leukemia	[166]
CP-4200	Acute Myeloid Leukemia, Breast Cancer, Colon Cancer	[167,168]
Gemcitabine	Cervical Cancer, Colorectal Cancer, Pancreatic Cancer, Bladder Cancer	[169,170,171,172,173,174]
Rx3117	Pancreatic Cancer, Bladder Cancer, Lung Cancer, Leukemic Lymphoblasts	[175,176]
Hydralazine	Prostate Cancer, Solid Cancers, Osteosarcoma	[177,178,179]

## 6. PARP1 Inhibitors in DNA Hypomethylation of Cancer Cells

### 6.1. PARP1 Inhibitors

The majority of PARP inhibitors are intended to challenge a binding site on the PARP1 molecule with nicotinamide adenine dinucleotide (NAD^+^) [180,181,182,183,184,185,186]. This approach led to the identification of NAD-like PARP inhibitors, which target not just PARP but also many other enzymatic pathways that use NAD^+^ and other nucleotides as co-factors [25,187,188,189]. Using certain inhibitors has a negative impact on several NAD^+^/nucleotide-dependent enzymatic pathways, resulting in additional deleterious consequences caused by the silencing of other pathways, while the PARP1 pathway efficiency is reduced. As a result, the approach is to implement inhibitors based on PARP1’s other functions [190,191,192].

Accumulating studies have suggested that the engagement of PARP1 with histone 4 (H4) caused by DSBs activates PARP1 enzymatic activity and enhances Alt-NHEJ [26,193,194,195,196]. To overcome this obstacle, researchers have devised a new technique for blindly screening a small chemical library for PARP1 inhibitors by focusing on a very specific mechanism of PARP1 activation [197]. A collection of PARP1 inhibitors has been chosen based on this screen, along with their structural categorization. The search found structurally unique non-NAD-like inhibitors that block PARP1’s role in cancer cells with better effectiveness and intensity than the conventional PARP1 inhibitors presently employed in treatments, besides drugs that exhibit structural similarities to NAD^+^ or existing PARP1 inhibitors., Identification of 5F02, a non-NAD-like inhibitor blocks the H4-mediated activity of PARP1 but not PARP2 or Tankyrase-1, and tested successfully against a variety of cancer cells, including BRCA1-deficient breast cancer line (MDA-MB-43) [198,199,200,201]. Non-NAD-like PARP1 inhibitors have shown effectiveness in targeting androgen-dependent and -independent pathways of androgen-receptor-signaling activation, in comparison to NAD-like PARP1 inhibitors. It has been experimentally revealed that the presence of esters and methylation of quaternary ammonium salt is crucial for 5F02′s anticancer action towards prostate-tumor growth [202]. In addition, researchers looked at the involvement of poly(ADP-ribose) glycohydrolase (PARG), a PARP1-related regulatory protein, in prostate carcinogenesis. According to the findings, *PARG* expression is significantly disrupted in prostate cancer cells (PC cells), which is linked to Cajal-body integrity and localization. Overall, the findings of our investigation support the use of non-NAD-like PARP1 inhibitors as a new therapeutic approach for progressive prostate cancer therapy [100]. PARP inhibitors use synthetic lethality to exploit homologous recombination (HR) deficiency and have emerged as potential anticancer medicines, particularly for BRCA1 or BRCA2 mutant carriers [203,204,205].

In Table 2, we summarize a list of significant PARP1 inhibitors, which are currently in clinical trials for different cancers, including prostate, breast, ovarian, liver cancers (solid) and lymphomas (non-Hodgkin).

### 6.2. PARP1 Inhibitors in Reversal of Tumor-Suppressor-Gene Expression

#### 6.2.1. Increase in DNA Hypomethylation by an Increase in TET Activity

PARP inhibitors ablate the poly(ADP-ribosyl)ation of TET. The DNA-hypermethylation-mediated silencing of TSGs is reversed by the inhibition of poly(ADP-ribosyl)ation activity of TET, thus enabling 5mC into 5HmC. Poly(ADP-ribosyl)ation stimulates TET1 enzymatic activity and TET1 activates PARP1 activity independently of DNA breaks (Figure 6) [101]. PARP activity positively regulates *TET1* expression by maintaining the DNA hypomethylation of CpG islands and H3K4 trimethylation [101]. TET1 is abundantly expressed in T-ALL cells and is required for in vivo human T-ALL cell proliferation. TET1 enzymatic capability to demethylate DNA, which permits it to retain global 5-hydroxymethylcytosine (5hmC) marks. To regulate leukemic development, controlling the cell cycle, DNA-repair genes and T-ALL-related oncogenes are prerequisites. PARP1 enzymes, which are correlated with increased expression in T-ALL patients, interact with the *TET*1 promoter to help create H3K4me^3^ modifications, thus accelerating transcription. *TET1* expression that is dependent on PARP1 might be inhibited by PARP1 inhibitors such as Olaparib, resulting in the removal of 5hmC marks, which could lead to the development of a therapy route for T-ALL cells [206].

#### 6.2.2. Maintenance of DNA Methylation by Poly(ADP-ribosyl)ation of CTCF and DNMT1

CTCF (CCCTC-binding factor) has been reported to be covalently poly(ADP-ribosyl)ated in vivo [207,208]. CTCF poly(ADP-ribosylation) inhibition stabilizes an upstream chromatin barrier and prevents neighboring heterochromatin from migrating into the active p16 tumor suppressor gene. In cancer development, the epigenetic inactivation of the *p16INK4a* tumor suppressor gene is a common target, which is an early marker in breast carcinogenesis. CTCF binds to this border and the absence of binding substantially correlates with p16 suppression in a variety of cancer cells [94,97]. CTCF binds to poly(ADP-ribosyl)ated PARP1 and DNMT1 unmethylated target sites, suggesting that PARP activity is essential to the maintenance of DNA methylation profiles. Loss of PARs, due to overexpression of *PARG*, results in the loss of CTCF and PARP1 DNA binding, as well as de novo methylation of CTCF-bound CpGs. These findings suggest that CTCF could contribute to the PARP-mediated safeguarding of certain DNA regions in their unmethylated form. Poly(ADP-ribosyl)ation, on the other hand, is accountable for maintaining the unmethylated condition of certain CTCF-bound CpGs. CTCF with PARP activity at its DNA target sites inhibits Dnmt1 functions, reducing de novo methylation of CpG dinucleotides. As a result of the de-repression of *DNMT1* by deficient PARP activity, CTCF DNA targets are hypermethylated [97,209,210].

## 7. Combination Therapy of DNA Methyltransferase Inhibitor and PARP Inhibitor

Cancers which are caused by compromised DNA-repair pathways are extremely sensitive to PARP1 inhibitors [116,211,212,213]. Acute myeloid leukemia (AML) is a heterogeneous cancer with a poor clinical prognosis. Previously, it has been reported that the *BRCA1* expression level is reduced in AML samples [214]. When AML is addressed with DNA-damaging drugs or radiation therapy, BRCA1 activity is lost, leading to the accumulation of genomic abnormalities and cancer cell death [215]. Novel combinations of DNMTs and PARP inhibitors could enhance effectiveness; for example, AML patients which are resistant to chemotherapy are treated with a novel combination therapeutic strategy that is more effective when combined with decitabine (DNMT inhibitors) [216]. In addition to demethylating CpG-island gene-promoter regions, it increases poly (ADP-ribose) polymerase (PARP1) interaction to DNA and strong association to chromatin, limiting PARP-mediated DNA-repair or -transcription activation, thus downregulating HR DNA repair and making cancer cells more sensitive to the PARP inhibitor [217]. In AML patients, high *PARP1* expression suggests poor survival. PARP inhibitors in association with histone-deacetylase inhibitors (SAHA–bendamustine hybrid) give a novel potential cure for AML. The combination effectively induces cell apoptosis and arrests the cancer cell cycle in the G2/M phase, thus delaying the development of AML and prolonging survival [206,218]. AML and acute lymphoblastic leukemia (ALL) have been both reported to have genomic alternations of PARP1 and compromised DNA-damage-response gene pathways. AML carries *RUNX1–RUNX1T1* (transcription factor involved in the differentiation of hematopoietic stem cells into adult blood cells) fusion genes with functional deficiency in *TET2* and *DNMT3A* genes [219], while *TET1* is highly sensitive to PARP1 inhibitors in ALL patients, as shown in different clinical studies [206]. By reversion of mutations in *BRCA1/2* genes, breast- and ovarian-cancer cells become resistant to PARP inhibitors [220,221,222]. Previous studies have demonstrated that DNMT inhibitors re-sensitize the resistance of breast or ovarian cells to PARP inhibitors, independent of BRCA1/2 mutation status [223]. Importantly, these combinations of DNMT and PARP inhibitors suggest that there is a budding and open pathway to develop a therapy for other cancers which are not completely dependent on gene mutations in the DNA-damage-repair pathway. The *TET2* gene is frequently mutated in malignant blood diseases (about ~50% in chronic myelomonocytic leukemia, CMML). In addition to demethylating genes, *TET2* has a significant role in DNA repair pathways, either single-strand breaks (SSB) or double-strand breaks (DSB). TET2 knock-down causes a decrease in *BRCA2* expression, which inhibits HR repair. In combination with a PARP inhibitor, both SSB and DSB are harmed, cell apoptosis is activated and cell survival is impeded [224].

## 8. Perspective and Conclusions

In recent studies on role of PARP1 in DNA dynamics, the extensive cross-talk between epigenetic pathways, including DNA methylation or demethylation and poly(ADP-ribosyl)ation of PARP1 itself or other target proteins, is evident. In cancer, both increases in and losses of DNA methylation are prevalent, but the processes that govern this methylation equilibrium are unknown. PARP1 plays a central role in DNA dynamics by enabling and executing DNMT1 or TET1 functions. Both proteins are functionally in contrast with each other, but their poly(ADP-ribosyl)ation depends on the enzymatical activation of PARP1 due to DNA damage in cancer cells or other cues. TET1 has been shown to trigger poly(ADP-ribosyl)ation independently of DNA damage to demethylate mouse primordial germ cells; this might be due to inhibitory DNMT1 activity or through the transcriptional up-regulation of the *TET1* gene [88].

DNMT1 is a major protein in the hypermethylation of cancer cells [225,226]. In different cancers, the degree of poly(ADP-ribosyl)ation of PARP1 varies; it also depends on the role of the related regulatory protein known as PARG1 that reverses the PARP1 function by removing poly(ADP)ribose moieties from target proteins [227,228,229]. In prostate cancer, poly(ADP-ribosyl)ation activity has been observed to increase several folds due to *PARG* expression being severally disrupted in these cells [100]. However, the DNA methylation status of TSGs and their expression in these cells are not known. Sirtuin gene activities are also disrupted in these cells because of NAD^+^ scarcity. It would be interesting to study the chromatin orchestration on TSGs and oncogenes in these cells with or without the use of PARP1 inhibitors. PARP1 inhibitors have been shown to be useful in the treatment of androgen-dependent malignancies [230,231,232].

Apart from inhibiting the enzymatic activity of DNMT1 by noncovalent poly(ADP-ribosyl)ation [87], poly(ADP-ribosyl)ated PARP1/ARTD1 positively controls DNMT1 expression [89]. Poly(ADP-ribosyl)ation of TET1 regulates transcriptional *TET1* expression [93]. In contrast to hypermethylation, TNBC is one of the most hypomethylated cancers [233]. TET1 DNA demethylase is notably overexpressed in roughly 40% of patients [234]. *PARP1* expression has been shown to be significantly increased in TNBC [235]. As a result, the US Food and Drug Administration (FDA) has approved two novel PARP inhibitors, Lynparza (olaparib) from AstraZeneca and Talzenna (talazoparib) from Pfizer, to manage metastatic TNBC patients with a BRCA (breast cancer type 1 susceptibility protein) mutation (or patients suspected to have one).

PARP1 positively regulates the transcription of genes by auto poly(ADP-ribosyl)ation and negatively by inhibiting the enzymatic activity of target proteins by covalent modification. In other words, PARP1, by binding to promoter regions, suppresses the transcription of genes, while auto-poly(ribosy)lated PARP1 activates transcription by providing access to transcription factors [82,236]. Despite the multiple functions of PARP1 in terms of transcription and DNA methylation/demethylation, mice have been shown to grow normally under *PARP1* knockout conditions [237], whereas it has been reported that double knockout *PARP1/PARP2* mice died in early embryogenesis [238]. This suggests that there is redundancy in the function of the *PARP* gene family, although PARP1 alone poly-ribosylates ~80–90% of target proteins [239].

DNA methylation is a dynamic and multi-regulated process; DNMT1 and TET1 are directly controlled and Poly(ADP-ribosyl)ation, PARP1, or PARG indirectly affects DNA methylation. In cancer, this tight regulation is disrupted by the increase or decrease in enzymatic activities and expression of these genes. In AML and breast cancer cells, DNA demethylating agents (DNMT inhibitors) improve the lethal action of PARP1 inhibitors [240,241,242,243]. Both inhibitors’ synergistic impact enhances DNA damage and, as a result, tumor cytotoxicity [244,245]. Notably, this combined treatment method improves the PARP1 inhibitor potency in cancer cells. Furthermore, it may be used in the future as part of combination therapy for cancers in which the DNA-damage-repair mechanism is disrupted. The transcription reactivation of TSG *p16* is indirectly regulated by poly(ADP-ribosyl)ation of CTCF and the lack of binding substantially correlates with *p16* silence in a wide range of cancers [97]. This merging of the multiple functions of proteins is the research area in which the epigenetic role of poly(ADP-ribosyl)ation has become increasingly evident; not only it controls gene transcription/expression but it also affects their biological functions by affecting their enzymatic activities and modifying the respective pathways. Research in this direction would aid in obtaining a mechanistic understanding for governing the epigenetic dynamic alterations which drive biological and cellular processes, including development and differentiation, and an increased prevalence of illnesses.

## Figures and Tables

**Figure 1 biomolecules-12-00417-f001:**
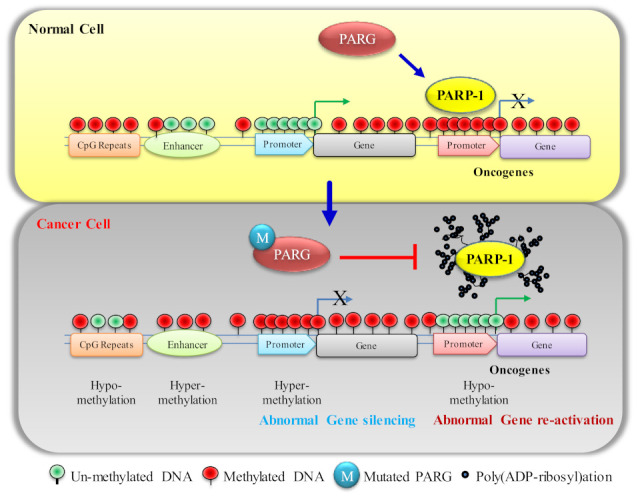
The difference in differential DNA methylation in normal and cancer cells. In PARG or homologous DNA-repair-defective cancer cells the enzymatic activity of PARP1 is increased; therefore, auto-poly(ADP-ribosy)lated PARP1 moves away from the pre-occupied promoter of oncogenes and provides the access to transcription machinery for expression. In downstream, it facilitates DNA hypomethylation at the promoters of oncogenes and make them transcriptional active De novo DNA methyltransferases (DNMTs) promote DNA methylation by catalyzing the transfer of a methyl group from donor S-adenosyl-l-methionine (SAM) to cytosine bases to produce 5mC. The DNMT family consists of five members—DNMT1, DNMT2, DNMT3A, DNMT3B and DNMT3L (Figure 2A) [35,36]. Interestingly, DNMTs’ important actions during DNA methylation may be divided into two categories, methylation maintenance and de novo methylation. DNA methylation is predominantly maintained by DNMT1, which facilitates copying DNA methylation patterns during DNA replication in the S phase of mitosis and meiosis [37]. The epigenetic mark can then self-replicate because of DNMT maintenance, which recognizes mono-methylation and methylates the CpG site’s complementary strand, leading to a di-methylated tag. Double-stranded methylation, during which the two methyl groups accept a *syn* conformation in the major groove, can modulate chromatin architecture and regulate gene transcription [38]. DNMT3A and DNMT3B are de novo methyltransferases; they potentially develop a new DNA methylation signature for unmethylated CpGs of DNA and are recognized as de novo DNMT enzymes [39,40]. DNMT3A or DNMT3B catalyzes the methylation of previously unmethylated DNA (de novo methylation) in embryonic stem cells and tumor cells [41]. DNMT3A and DNMT3B can also aid in the maintenance of DNA methylation [42,43]. Accumulating evidence suggests that DNMT3L (DNMT3-like) has no catalytic activity because some crucial motifs have been lost or altered [44]. However, DNMT3L contributes as an essential cofactor for de novo methyltransferase by expediting the interaction among DNMT3A, DNMT3B and DNA, and stimulating their activity [39,40,45,46].

**Figure 2 biomolecules-12-00417-f002:**
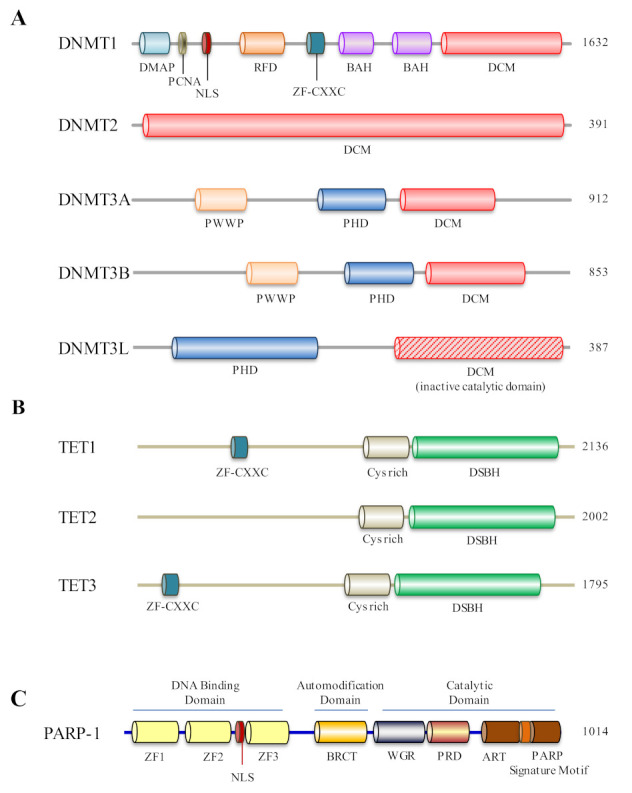
Schematic structure of human DNMT, DNMT3-like, TET family and PARP1 proteins. (**A**) DMAP, DMAP1-binding domain; PCNA, Proliferating Cell Nuclear Antigen domain; NLS, Nuclear Localization Signal domain; DNMT1-RFD, Cytosine-specific DNA methyltransferase replication foci domain; Zf-CXXC, CXXC zinc finger domain; BAH, Bromo adjacent homology domain; DCM, DNA-cytosine methylase; Cyt_C5_DNA_methylase, Cytosine-C5 specific DNA methylases; PWWP, domain comprising a conserved proline–tryptophan–tryptophan–proline motif; PHD, plant homeodomain; The sequences are derived from data reported under accession numbers NP_001124295 for DNMT1, NP_004403 for DNMT2, NP_783328 for DNMT3A, NP_008823 for DNMT3B and NP_037501 for DNMT3L. (**B**) Domain structures of ten–eleven translocation methylcytosine dioxygenases (TETs). Schematic representation of conserved domains of human TET proteins is shown, including a double-stranded-helix (DSBH) fold (all TETs), cysteine-rich (Cys-rich) domain (all TETs) and CXXC zinc fingers (Zf-CXXC; in TET1 and TET3). The sequences are derived from data reported under accession numbers NP_085128 for TET1, NP_001120680 for TET2 and NP_001274420 for TET3. (**C**) Domain structure of PARP1. PARP1 has four main domains, an amino (N)-terminal DNA-binding domain, an auto-modification domain, a water-binding domain and a carboxy (C)-terminal catalytic domain. ZFI, zinc finger I; ZF2, zinc finger II; ZF3, zinc finger III; NLS, nuclear localization signal; BRCT, BRCA1 C-terminal domain; PRD, PARP regulatory domain; ART, ADP-ribosyl transferase subdomain.

**Figure 3 biomolecules-12-00417-f003:**
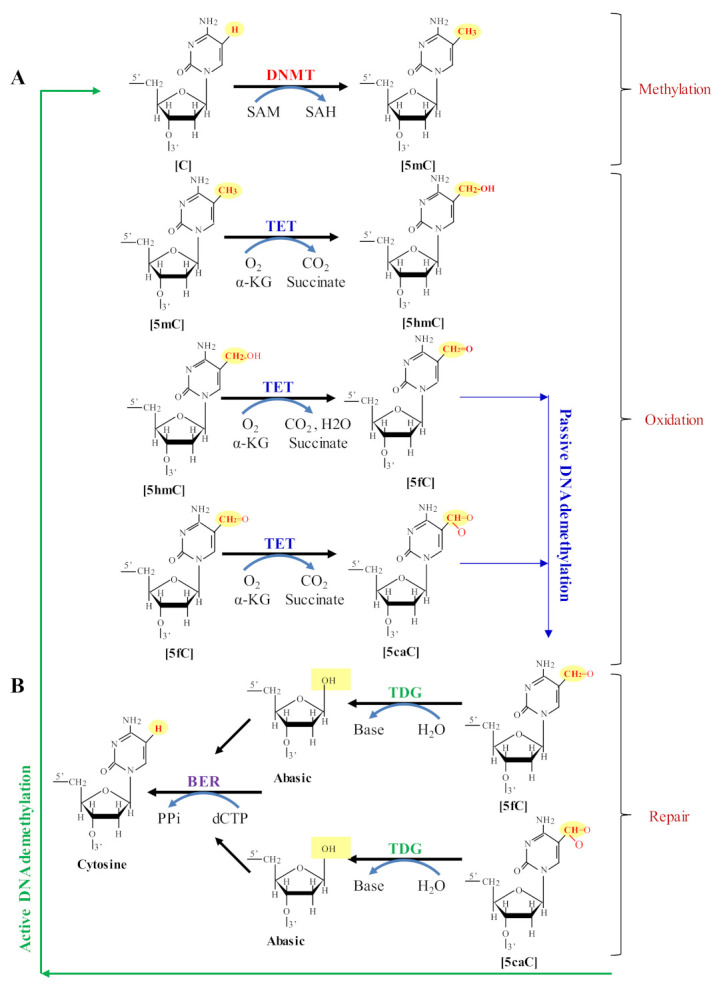
Steps for dynamic modifications of Cytosine and TET-mediated oxidation. (**A**) The methylation of deoxycytosine (C) residues to 5-methylcytosine (5mC) are introduced by DNA methyltransferase (DNMT) enzymes and sequentially oxidized by ten–eleven translocation (TET) enzymes via 5-hydroxymethylcytosine (5mC), 5-formylcytosine (5fC) and 5-carboxylcytosine (5caC). SAM, S-adenosylmethionine; SAH, S-adenosylhomocysteine; α-KG, α-ketoglutarate. (**B**) 5fC and 5caC are identified and excised by thymine DNA glycosylase (TDG) to produce an abasic site. The base-excision-repair (BER) pathway implicates excision of the abasic site, replacement of the nucleotide using unmodified deoxycytidine triphosphate (dCTP) by a DNA polymerase (generating pyrophosphate, PPi) and ligation to repair the nick.

**Figure 4 biomolecules-12-00417-f004:**
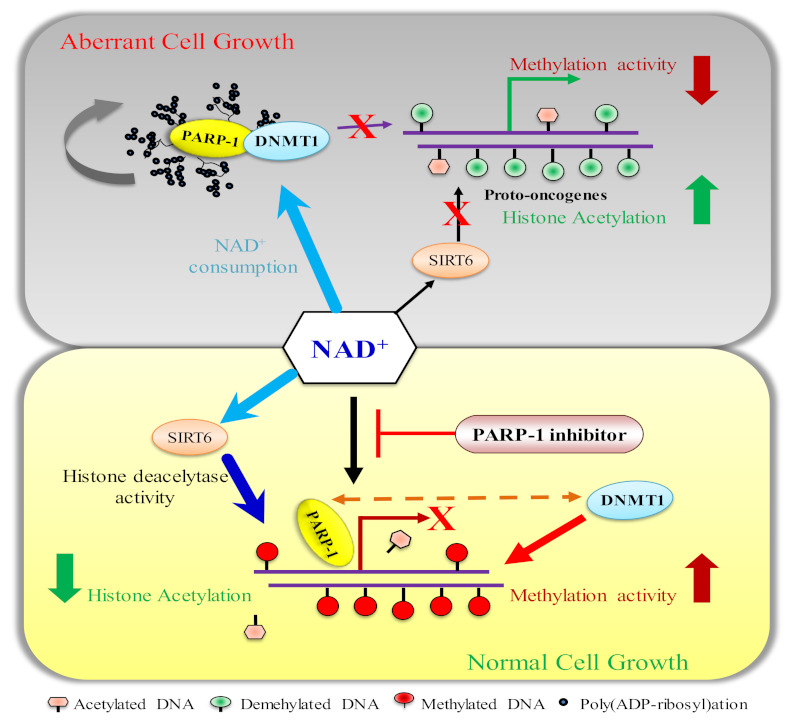
Increased poly(ADP-ribosyl)ation precludes DNMT1 and SIRT6 enzymatic activities. In cancer cells (prostate), the poly(ADP-ribosyl)ation pathway is severely disrupted, resulting in an enhanced activity that not only poly(ADP-ribosyl)ates PARP1 but also DNMT1; therefore, it prevents the maintenance of DNA methylation on newly synthesized DNA strands. Scarcity of NAD^+^ makes SIRT6 enzymatically inactive to remove the acetyl group from histone proteins, eventually facilitating the transcription of oncogenes. The PARP1 inhibitor reverts all activities, which leads to the suppression of oncogenes.

**Figure 5 biomolecules-12-00417-f005:**
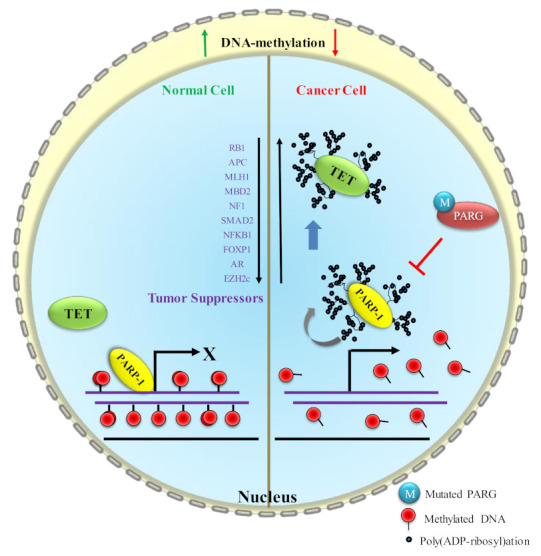
Enhanced poly(ADP-ribosyl)ation maintains DNA hypomethylation by activating TET1 functions. PARP1 binds to TSGs in their promoter region in normal cells. In PARG or homologous DNA-repair-defective cells, the enzymatic activity of PARP1 is increased (although TET1 activates PARP1 independently of DNA breaks) and poly(ADP-ribosyl)ated TET1 performs its DNA de-methylation function. It leads DNA hypomethylation on the regulatory regions of genes in cancer cells. Eventually, it facilitates the expression of TSGs; although this is not quite straightforward in malignant cells, it is a model to understand the functional dependency of proteins to each other and it is helpful to develop therapies by taking advantage of them.

**Figure 6 biomolecules-12-00417-f006:**
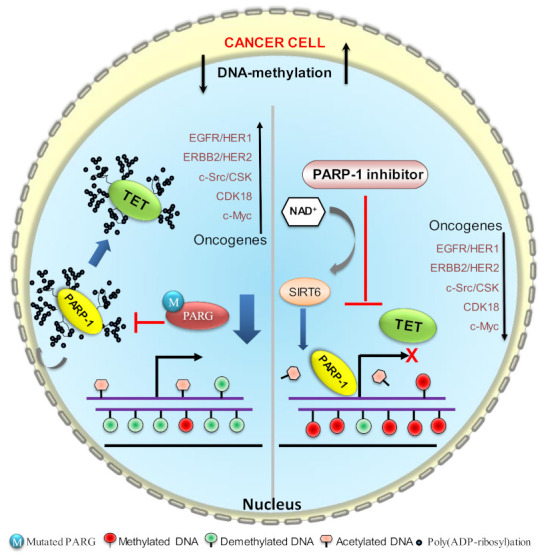
TET1 stimulates the activity of PARP1 independently of DNA damage. Poly(ADP-ribosyl)ation of TET1 by PARP1 increases TET1 enzymatic activity and regulates the hydroxylase activity of the DNA demethylation processes. Poly(ADP-ribosyl)ation of TET1 preserves the unmethylated state and activates the transcription of oncogenes. The PARP1 inhibitor inhibits the enzymatic activity of TET1, as a result, the expression of oncogenes is downregulated due to hypermethylation. Availability of NAD^+^ makes SIRT6 enzymatically active to remove the acetyl group from histone proteins, further downregulating the transcription of oncogenes.

**Table 2 biomolecules-12-00417-t002:** Clinical trials of PARP1 inhibitors.

PARP1 Inhibitor	Cancer Type	NCT Number *
Lynparza/Olaparib	Ovarian Cancer Breast Cancer	NCT04041128 NCT04826198 NCT04774406 NCT03462342 NCT04065269 NCT03150576 NCT04582552 NCT04774406
Cyh33	Ovarian Cancer Breast Cancer Solid Tumor Prostate Cancer Endometrial Cancer	NCT04586335
Talazoparib	Neuroendocrine Tumors	NCT05053854
Rp12146	Solid Tumor Lung Cancer Breast Cancer Ovarian Cancer	NCT05002868
Niraparib	Advanced Solid Tumors (Excluding Prostate Cancer) Ovarian Cancer Head And Neck Squamous Cell Carcinoma	NCT04267939 NCT04826198 NCT04774406 NCT04734665 NCT04681469 NCT04837209 NCT04774406
Idx-1197	Solid Tumors	NCT04174716
Talazoparib	Breast Cancer	NCT03990896 NCT04774406
Rucaparib	Solid Tumor	NCT04276376 NCT04774406
Veliparib	Solid Tumors Liver Tumors Lymphomas Prostate Cancer	NCT01434316 NCT01618357

* These data were obtained from https://www.clinicaltrials.gov/ (accessed on 30 December 2021). NCT, National Clinical Trial number.

## Data Availability

Not applicable.

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
