# Peer review of "DNA Methylation Malleability and Dysregulation in Cancer Progression: Understanding the Role of PARP1"

_biomolecules, 2022, doi:10.3390/biom12030417_

Round 1

Reviewer 1 Report

The review article “DNA Methylation Malleability and Dysregulation in Cancer Progression: Understanding PARP-1’s Role” by Srivastava and Lodhi summarizes and discusses a wide range of literature about DNA methylation and the relevance of its aberrations in carcinogenesis.

The Authors emphasize how complex the role of DNA methylation in cancer is referring to several sets of results. The review is not entirely novel. It describes well-known mechanisms of dysregulated DNA methylation in cancer and the role of DNMTs, TETs and PARP1 proteins in tumor development. Over 75% of references are older than 2016. Although the Authors provide some information supported by abundant literature, the organization and clarity of the data should be improved and updated, especially the parts on DNMTs and PARP1 inhibitors. Enhancing the cytotoxic effects of PARP inhibitors with DNA demethylating agents could be discussed (DOI: 10.1016/j.ccell.2016.09.002).

English editing is required to improve the article for language and style. Many sentences are too long and not clear. It makes the paper hard to read. Example: “Passive DNA demethylation, on either hand, occurs as a result of the loss of maintenance methylation induced by multiple cycles of replication without the functioning of DNA methylation repair machinery and also can proceed during mammalian development, for instance, DNMT1 suppression or absence of DNA hypomethylation impact and also in pre-implantation development in the maternal genome [43,44].”

Figure legends should be more clear and consistent, including an explanation of all abbreviations depicted on the Figures. Figure 2 showing the well-known structure of DNMTs, TETs and PARP1 proteins is not necessary.

Minor comments:

The abbreviation list should be re-checked and corrected – some abbreviations are explained in the text of the paper but many are not (TDG, MBD4, PDAC, CTCF, DSBs, etc.). Please explain abbreviations when they are used for the first time in the text. All the abbreviations should be used consistently throughout the manuscript (e.g. there are two different abbreviations for tumor suppressor gene: TSG and TSP gene, etc.).

Names of the genes should be written in italic (e.g. “the DNMT1 gene”, page 10, line 279, etc.).

Author Response

Review Report (Reviewer 1):

We are very thankful to reviewer for his/her constructive comments and suggestions in the entire review process.  
  1. Out of 164 references, there are only 3 references published in 2021, 4 in 2020 and 12 in 2019, which means many up-to-date discoveries on the PARP1, chromatin remodeling and epigenetic regulation are not included in this review. In Pubmed, the key words “PARP1, Chromatin” are related to 70 publications in 2021 and 52 in 2020. “PARP1, DNA methylation” are related to 34 publication from 2020 to 2021.

Response: We are very thankful to the reviewer for constructive comments and suggestions in the entire review process. We agree with the reviewer to incorporate recent published research work and references (now 249 numbers) in the revised manuscript accordingly. We hope that our revision satisfies the reviewer and matches the standard to publish in the Biomolecules journal.

  1. Some basic concept of PARylation is not precise. The author stated “When chromatin-associated PARP-1 is activated, ADP-ribose is transferred to form poly(ADP-ribose) (pADPr) to glutamate, aspartate, lysine, arginine, and serine residues of various substrates including existing pADPr chain and PARP-1 itself characterized by poly(ADP-ribosyl)ation [96] “. Actually, the PARylation starts as directly transferring ADP-ribosyl group from NAD+ to target proteins by PARP1/2 and then immediately the PAR chain can be elongated linearly or branched by adding additional PAR units (PMID: 33804735). This reaction is continuous other than “adding PAR in existing pADPr chain”

Response: We agree with the reviewer and thank you so much for this nice suggestion. We modified the sentence carefully and corrected it. It will improve the quality of our manuscript. We removed overlapped information and included the mentioned reference.

  1. In Figure 4 and 5, the stable PARylated of DNMT1 is impossible due to the activity of PARG. PARG is the most important protein to degrade PAR chain very fast to maintain the homeostasis between PARylation and de-PARylation in either normal or cancer cells. The PAR is almost undetectable in cells without DNA damage treatment due to the quick balance between the activity of PARP1/2 and PARG. Even with DNA damage treatment, PAR signal disappears quickly after reaches the peak (usually 15~30min after MNNG/MMS/H2O2 treatment)

Response: We thank the reviewer for him/her raising this point, we agree that PARylation is reversible PARP1 enzymatic activity and PARG quickly maintains the homeostasis in the cell by degrading the PAR chain that prevents the disastrous effect on the cell. However, previously published papers (PMID: 19262751, PMID: 15637587, PMID: 26136340) show PARylated PARP1 interacted with DNMT1 and form complex and prevents DNMT1 functional activity to methylate DNA. Accordingly, we corrected Fig. 4 DNMT1 is not PARylated instead auto-PARylated PARP-1 binds to DNMT1 and prevents its function. TET1 activates PARP1 independently of DNA breaks (PMID: 26136340) and our lab reported that interaction with N- terminal tail of histone H4 triggers PARP1 activity (PMID: 17827147). In PARG or homologs DNA repair defective cells, the enzymatic activity of PARP1 is increased and imbalance the NAD homeostasis (PMID: 20551068, PMID: 30880062). We hope now it will help to understand better for readers after adding the description in the revised manuscript.

  1. The domains of PARP1 were not presented correctly. There are much more domains in PARP1 other the single bar in this manuscript. There is a nice scheme of PARP1 in the paper with PMID: 31842403.

Response: We thank the reviewer to attract our attention to an important part of the review. We agree with the suggestion and updated Figure 2C and described functions of different PARP1 domains in the revised manuscript. We included suggested references.

Kind regards,

Reviewer 2 Report

Out of 164 references, there are only 3 references published in 2021, 4 in 2020 and 12 in 2019, which means many up-to-date discoveries on the PARP1, chromatin remodeling and epigenetic regulation are not included in this review. In Pubmed, the key words “PARP1, Chromatin” are related to 70 publications in 2021 and 52 in 2020. “PARP1, DNA methylation” are related to 34 publication from 2020 to 2021.

Some basic concept of PARylation is not precise. The author stated “When chromatin-associated PARP-1 is activated, ADP-ribose is transferred to form poly(ADP-ribose) (pADPr) to glutamate, aspartate, lysine, arginine, and serine residues of various substrates including existing pADPr chain and PARP-1 itself characterized by poly(ADP-ribosyl)ation [96] “. Actually, the PARylation starts as directly transferring ADP-ribosyl group from NAD+ to target proteins by PARP1/2 and then immediately the PAR chain can be elongated linearly or branched by adding additional PAR units (PMID: 33804735). This reaction is continuous other than “adding PAR in existing pADPr chain”     

In Figure 4 and 5, the stable PARylated of DNMT1 is impossible due to the activity of PARG. PARG is the most important protein to degrade PAR chain very fast to maintain the homeostasis between PARylation and de-PARylation in either normal or cancer cells. The PAR is almost undetectable in cells without DNA damage treatment due to the quick balance between the activity of PARP1/2 and PARG. Even with DNA damage treatment, PAR signal disappears quickly after reaches the peak (usually 15~30min after MNNG/MMS/H2O2 treatment)

The domains of PARP1 were not presented correctly. There are much more domains in PARP1 other the single bar in this manuscript. There is a nice scheme of PARP1 in the paper with PMID: 31842403.

Author Response

Review Report (Reviewer 2):

We are very thankful to reviewer for his/her constructive comments and suggestions in the entire review process.  

The review article “DNA Methylation Malleability and Dysregulation in Cancer Progression: Understanding PARP-1’s Role” by Srivastava and Lodhi summarizes and discusses a wide range of literature about DNA methylation and the relevance of its aberrations in carcinogenesis. The Authors emphasize how complex the role of DNA methylation in cancer is referring to several sets of results. The review is not entirely novel. It describes well-known mechanisms of dysregulated DNA methylation in cancer and the role of DNMTs, TETs, and PARP1 proteins in tumor development.

  1. Over 75% of references are older than 2016. Although the Authors provide some information supported by abundant literature, the organization and clarity of the data should be improved and updated, especially the parts on DNMTs and PARP1 inhibitors. Enhancing the cytotoxic effects of PARP inhibitors with DNA demethylating agents could be discussed (DOI: 10.1016/j.ccell.2016.09.002).

Response: Thank you so much for your critical suggestions. We have modified the manuscript and updated the new recent research published papers. We have added two tables for giving the recent information about DNMTs and PARP1 inhibitors. We have also added information about combined therapy for DNMTs and PARP1 inhibitors for regulating cancer progression. We also included the reference provided by the reviewer accordingly to the text. We anticipate that our revision fulfills the reviewer’s concern and improved the quality of the manuscript.  

  1. English editing is required to improve the article for language and style. Many sentences are too long and not clear. It makes the paper hard to read. Example: “Passive DNA demethylation, on either hand, occurs as a result of the loss of maintenance methylation induced by multiple cycles of replication without the functioning of DNA methylation repair machinery and also can proceed during mammalian development, for instance, DNMT1 suppression or absence of DNA hypomethylation impact and also in pre-implantation development in the maternal genome [43,44].”

Response: We express regret for the inconvenience, completely agree with the reviewer, and thank him/her for the suggestion. We have reviewed the manuscript for English editing, revised the wordings carefully to remove the overstatements, and reframe them into small sentences for easy understanding.

  1. Figure legends should be more clear and consistent, including an explanation of all abbreviations depicted on the Figures. Figure 2 shows the well-known structure of DNMTs, TETs, and PARP1 proteins are not necessary.

Response: We modified the figure legend included an explanation of abbreviations of figure legends. About Figure 2, we agree with the reviewer, however, we think in context description in manuscript inclusion of figure will help to understand better different functions of domains of DNMTs, TETs, and PARP-1 to readers. The figures are also modified according to the reviewer's suggestions.

Minor comments:

  1. The abbreviation list should be re-checked and corrected – some abbreviations are explained in the text of the paper but many are not (TDG, MBD4, PDAC, CTCF, DSBs, etc). Please explain abbreviations when they are used for the first time in the text. All the abbreviations should be used consistently throughout the manuscript (e.g. there are two different abbreviations for tumor suppressor gene: TSG and TSP gene, etc.). Names of the genes should be written in italic (e.g. “the DNMT1 gene”, page 10, line 279, etc.).

Response: We agree with the reviewer and thank for the suggestion. In the revised version, we included an abbreviation for protein or gene each for better understanding to readers and all gene’s names are italicized. Thank you for your critical comments to improve the quality of manuscript.

Kind regards,

Round 2

Reviewer 1 Report

The Authors have satisfactorily responded to all my comments and made the necessary changes to the manuscript.

Minor comment: In the Figure 1 legend (page 3, lines 96-98), the following sentence "The presence of DNA hypomethylation leads to the suppression expression of tumor suppressor oncogenes genes." should be corrected.

Reviewer 2 Report

The manuscript is much better after revision. I would recommend accepting it for publication.